# Improving the Structural Parameter of the Membrane Sublayer for Enhanced Forward Osmosis

**DOI:** 10.3390/membranes11060448

**Published:** 2021-06-15

**Authors:** Jin Fei Sark, Nora Jullok, Woei Jye Lau

**Affiliations:** 1Faculty of Chemical Engineering Technology, Universiti Malaysia Perlis, Kompleks Pusat Pengajian Jejawi 3, Arau 02600, Perlis, Malaysia; skysark19@gmail.com; 2Centre of Excellence for Biomass Utilization, Universiti Malaysia Perlis, Kompleks Pusat Pengajian Jejawi 3, Arau 02600, Perlis, Malaysia; 3Advanced Membrane Technology Research Centre, Universiti Teknologi Malaysia, Skudai 81310, Johor, Malaysia; lwoeijye@utm.my

**Keywords:** structural parameter, forward osmosis, composite membrane, sublayer

## Abstract

The structural (*S*) parameter of a medium is used to represent the mass transport resistance of an asymmetric membrane. In this study, we aimed to fabricate a membrane sublayer using a novel composition to improve the *S* parameter for enhanced forward osmosis (FO). Thin film composite (TFC) membranes using polyamide (PA) as an active layer and different polysulfone:polyethersulfone (PSf:PES) supports as sublayers were prepared via the phase inversion technique, followed by interfacial polymerization. The membrane made with a PSf:PES ratio of 2:3 was observed to have the lowest contact angle (CA) with the highest overall porosity. It also had the highest water permeability (*A*; 3.79 ± 1.06 L m^−2^ h^−1^ bar^−1^) and salt permeability (*B*; 8.42 ± 2.34 g m^−2^ h^−1^), as well as a good NaCl rejection rate of 74%. An increase in porosity at elevated temperatures from 30 to 40 °C decreased *S_int_* from 184 ± 4 to 159 ± 2 μm. At elevated temperatures, significant increases in the water flux from 13.81 to 42.86 L m^−2^ h^−1^ and reverse salt flux (RSF) from 12.74 to 460 g m^−2^ h^−1^ occur, reducing *S_eff_* from 152 ± 26 to 120 ± 14 μm. *S_int_* is a temperature-dependent parameter, whereas *S_eff_* can only be reduced in a high-water- permeability membrane at elevated temperatures.

## 1. Introduction

Forward osmosis (FO) is the spontaneous movement of water molecules from a low-concentration feed solution (FS) to a high-concentration draw solution (DS) based on the osmotic pressure gradient across a semi-permeable membrane without the application of external hydraulic pressure [1,2]. Thin film composite (TFC) membranes, which exhibit good water permeability and excellent solute rejection, are the most common type of membrane used in FO processes [3]. Their development was inspired by TFC reverse osmosis (RO) membranes [4], which typically consist of thick, non-woven fabric and a polymeric support that creates drastic internal concentration polarization (ICP) [5]. These TFC membranes were redesigned without the non-woven fabric for FO applications with the aim of increasing membrane performances [6,7]. Ideally, thin film composite forward osmosis (TFC FO) membranes should exhibit high porosity, low tortuosity, and low membrane thickness, which together contribute to high water permeability (*A*), low solute permeability (*B*), and low values of the structural parameter (*S*) [3,6,8].

The *S* parameter of a TFC FO membrane has long been used to measure the degree of ICP in osmotically-driven processes [6,8,9,10]. At an early stage, the *S* parameter is called the intrinsic structural parameter or *S_int_*, and it is determined by measuring the geometric variables (i.e., membrane porosity, tortuosity, and thickness). This parameter value is constant and remains unaffected by operating conditions [2,6,11,12]. However, this model is questionable and challengeable and should be investigated by varying the operating pressure and temperature [2,13]. Although characterizations of the membrane porosity and tortuosity were carried out successfully by Manickam et al. [14], the resulting *S_int_* is not accurate under real FO conditions in which the temperature and pressure are not consistent throughout the process.

Afterwards, a fitted parametric model utilizing the effective structural parameter, *S_eff_* was developed and adopted by most researchers [4,8]. *S_eff_* is fundamentally different from *S_int_* in that it is obtained by measuring experimental data, although both parameters are used to determine the degree of ICP [8]. The *S_eff_* model is also questionable because its values cannot express the intrinsic membrane properties and lack of specifications unless the operating conditions are standardized [14,15,16]. However, various studies have reported noticeable deviations in *S_eff_* even under the same operating conditions and when using identical membrane samples [2,4,16,17,18]. Thus, the inaccuracies in the model are still the main concern [8].

Kang et al. [8] developed a pore-scale simulation *S_eff_* model by considering interfacial porosity and partially correlated the discrepancy between *S_int_* and *S_eff_*. Their proposed model using *S_eff_* still deviates significantly from real FO operations. Abdelkader and Sharqawy [19] correlated the *S* parameter of the membrane with temperature by constant membrane properties. Little effort has been made with regards to the interaction between the membrane morphology-solution and temperature, even though it is believed to play a significant role in influencing the *S_int_* [2]. An evaluation of the responses of membrane thickness and tortuosity to temperature is difficult to carry out, but it remains certain that the water temperature can also induce changes in membrane porosity that can be measured, as shown in the present study. The response of the membrane morphology to temperature, which contributes directly to the *S_int_* value, is worthy of further investigation.

Hence, the main objective of this work is to investigate the effects of substrates made with different polysulfone (PSf) and polyethersulfone (PES) blend ratios on the *S* parameter at elevated temperatures in FO operations using laboratory-synthesized TFC FO membranes. The PSf:PES substrates used in Sun et al. [20] were evaluated in both RO and FO experiments, and gravimetry measurements were taken to obtain overall membrane porosities at different temperatures. Three different temperatures were used to evaluate the changes in *S_eff_* and *S_int_*, and these changes were later compared to the expected deviations between *S_int_* and *S_eff_* arising from the computation models.

## 2. Modelling of Solution Hydrodynamics

The osmotic pressure of a solution is obtained using Van’t Hoff’s equation [10]:(1)π=iCRT
where *i* is the Van’t Hoff factor (it is 2 for NaCl), *C* is the solute concentration (mol L^−1^), *T* is the absolute temperature (K), and *R* is the gas constant (0.08314 L bar K^−1^ mol^−1^). In an osmotically-driven process, concentration polarization is one of the severe problems that decreases the effective osmotic driving force [8]. In a FO process (Figure 1), two different types of concentration polarization occur on the membrane surface and within the membrane substrate; namely, concentrative external concentration polarization (cECP) and dilutive internal concentration polarization (dICP), respectively [21].

The cECP is caused by the impurities in the FS becoming concentrated on the membrane surface via the convective water pull force [21]. This can be derived from film theory using Equation (2):(2)$$\frac{\pi_{F,m}}{\pi_{F,b}}=\exp\left(\frac{J_{w}}{k}\right)$$
where $\pi_{F,m}$ (bar) is the osmotic pressure at the membrane surface, and $\pi_{F,b}$ (bar) is the osmotic pressure of the bulk DS. Note that the water flux, $J_{W}$ (L m^−2^ h^−1^) is positive, indicating that the direction of the water flux is predicted to be from the FS to the DS and that $\pi_{F,m}>\pi_{F,b}$ due to the salt concentrated on the membrane surface. The water flux, *J_w_* for an osmotically-driven membrane is defined as [22]:(3)$$J_{w}=A\left(\pi_{D,b}-\pi_{F,b}\right)$$
where *A* is the water permeability coefficient, and $\pi_{D,b}$ is the bulk osmotic pressure of the DS. Meanwhile, the mass transfer coefficient is described as:(4)$$k=\frac{Sh\;D}{d_{h}}$$
where *D* (m^2^ s^−1^) is the diffusion coefficient of the salts based on the temperature of the DS, and *d_h_* (m) is the hydraulic diameter of the flow channel [23]. According to Touati and Tadeo [24], the Sherwood number (*Sh*) can be calculated for the appropriate flow regime in a rectangular channel using the following equations:(5)$$Sh=1.85{\left(Re\;Sc\frac{d_{h}}{X}\right)}^{0.33}\;(\mathrm{laminar}\;\mathrm{flow})$$
(6)$$Sh=0.04Re^{0.75}Sc^{0.33}\;\left(\mathrm{turbulent}\;\mathrm{flow}\right)$$
where *X* (m) is the length of flow channel, *Re* is Reynolds number calculated from Equation (7) below, and *Sc* is the Schmidt number obtained from Equation (8):(7)$$Re=\frac{v\;d}{\mu }$$
(8)$$Sc=\frac{\mu }{D}$$
where *v* (m s^−1^) is the flow velocity in the channel, *d* (m) is the channel diameter, and *μ* (m^2^ s^−1^) is the kinematic viscosity of the solution. An empirical equation can be used to estimate the kinematic viscosity:(9)$$\frac{\mu }{\mu_{w}}=1+0.12C\exp\left(\frac{C^{0.44}}{3.713T_{R}+2.792}\right)$$
where $\mu_{w}$ (m^2^ s^−1^) is the kinematic viscosity of water at different temperatures, *C* (mol L^−1^) is the molar concentration of the DS, and $T_{R}$ is normalized temperature calculated with:(10)$$T_{R}=\frac{T}{273.15}$$

The dICP occurs within the membrane substrate as a result of high water permeation and reduced effective osmotic pressure (Figure 1) and can be expressed as:(11)$$\frac{\pi_{D,m}}{\pi_{D,b}}=\exp\left(-J_{w}K\right)$$
where $\pi_{D,m}$ is the osmotic pressure of the DS on the membrane surface that expresses the salt accumulation (when it is negative, the slat is being transported in the direction opposite to the water flux), and *K* is the mass resistivity. The mass resistivity *K* for diffusion within the membrane substrate is defined as [24]:(12)$$K=\frac{\tau t}{D\varepsilon }=\frac{S}{D}$$
where *τ* is the tortuosity, *t* is the membrane thickness, *ε* is the porosity, and *D* is the diffusion coefficient. Equation (13) is used to describe the ICP model for mass transfer resistance developed by McCutcheon and Elimelech [21]:(13)$$K=(\frac{1}{J_{w}})\ln\frac{B+A\pi_{D,b}}{B+J_{w}+A\pi_{F,m}}$$

Based on the Stokes–Einstein relationship, *D* can be obtained from:(14)$$D=\frac{\beta T}{6\pi r\rho \mu }$$
where *β* is the Boltzmann constant, *T* (°C) is the absolute temperature, *π* (bar) is the bulk osmotic pressure of the DS, *r* (m) is the ion radius, *ρ* (kg m^−3^) is the density of the DS, and *μ* (m^2^ s^−1^) is the kinematic viscosity of the DS. The dynamic viscosity (kg m^−1^ s^−1^) of the NaCl solution is obtained with [24]:(15)$$\eta \left(T\right)=2.414\times 10^{\left(\frac{247.8}{T-140}-5\right)}$$

The water flux equation, which considers the effects of cECP and dICP, is described as:(16)$$J_{w}=(\frac{1}{K})\ln\frac{B+A\pi_{D,b}}{B+J_{w}+A\pi_{F,b}\exp\left(J_{w}/k\right)}$$

The structural parameter *S* is the measure of the actual path length of the water diffusion across the support membrane or the degree of ICP [4,25]. Generally, *S_int_* is directly proportional to the membrane thickness and tortuosity and inversely proportional to porosity, which represents the membrane morphology [9]. The *S_int_* (μm) is defined as:(17)$$S_{int}=\frac{\tau \;t}{\varepsilon }$$
where *τ* is the dimensionless membrane tortuosity (1 < *τ* < 2), ε (%) is the overall membrane porosity, and *t* (μm) is the membrane thickness. *S_eff_* is computed from experimental data for the FO mode:(18)$$S_{eff,FO}=\frac{D}{J_{w}}\ln\frac{\pi_{D}\exp\left(J_{w}/k\right)+\frac{J_{w}-B}{A}}{\pi_{F}+\frac{B}{A}}$$
where *D* (m^2^ s^−1^) is the salt diffusion coefficient from Equation (14), *J_w_* (LMH) is the experimental water flux. A (L m^−2^ h^−1^ bar^−1^) and B (g m^−2^ h^−1^) are obtained from Equations (21) and (23), respectively [10].

## 3. Methodology

### 3.1. Materials

Polysulfone (PSf, Udel^®^ P-1700, MW = 67,000 g mol^−1^, Solvay) and polyethersulfone (PES, 3000 P, MW = 58,000 g mol^−1^, Merck) were dried overnight in an oven at 80 °C. N-methyl-2-pyrrolidone (NMP, ≥99.5%, MW = 99.13 g mol^−1^), polyethylene glycol 600 (PEG 600, MW = 570 g mol^−1^), 1,4-phenylenediamine (PPD, ≥99%, MW = 108.14 g mol^−1^), 1,3,5-benzenetricarbonyl trichloride (TMC, 98%, MW = 265.47 g mol^−1^), n-hexane (ACS grade, ≥96%, MW = 86.18 g mol^−1^) were all purchased from Merck & Co., Darmstadt, Germany. The sodium chloride (NaCl, MW = 58.44 g mol^−1^) was purchased from Hamburg Chemicals, Hamburg, Germany.

### 3.2. Experimental Procedure

#### 3.2.1. Fabrication of the Membrane Support Layer

Dope solutions containing mixtures of PSf:PES with different weight ratios (5:0, 2:3, and 0:5) were prepared and labelled as S1, S2, and S3, respectively as shown in Table 1. The polymer dopes were blended continuously on a low-profile roller to obtain homogenous solutions. The dope solutions were then stored in a vacuum oven for 24 h to remove any trace of air bubbles. Each dope solution was cast onto a glass plate using a casting knife with a thickness of 200 µm. The glass plate was then immersed in a deionized water bath to allow the phase inversion process to take place. A white film was produced fully in the water bath. The precipitated support membrane was kept in the deionized water bath for 24 h to ensure the removal of any remaining solvent in the membrane.

#### 3.2.2. Interfacial Polymerization of the TFC FO Membranes

Each support membrane was immersed in an aqueous solution containing 2 wt.% PPD for 3 min. The excess PPD solution was poured off and removed by filter paper, and the support membrane was allowed to dry for 20 s. Then 0.1 wt.% of TMC in n-hexane was poured onto the support membrane for 1 min to permit the interfacial polymerization process to form a PA active layer on the membrane’s top surface. The resulting TFC FO membranes were labelled MS1, MS2, and MS3. They were kept in the oven at 60 °C for 30 min, then air cooled for 2 min. Finally, the thin film composite forward osmosis (TFC FO) membranes were stored in deionized water at room temperature prior to usage.

### 3.3. Characterization of the TFC FO Membranes

#### 3.3.1. Scanning Electron Microscopy (SEM)

SEM (JEOL JSM-6010LV, Tokyo, Japan) was performed to investigate the top surfaces and cross-sectional structures of the TFC FO membranes. To obtain the cross sections of the membranes, the membranes were freeze-dried and then broken up in liquid nitrogen. All the membrane samples were sputtered with platinum under vacuum conditions so that they could conduct electricity during the SEM analyses.

#### 3.3.2. Contact angle (CA) Measurements

The wettability analyses of the synthesized membranes were done via CA measurements using an OCA–15Pro from Dataphysics (Filderstadt, Germany). Samples of membranes were air dried at 25 ± 1 °C for 24 h prior to testing. The average value of the water CA for each membrane sample was obtained by averaging six measurements to minimize experimental errors.

#### 3.3.3. Porosity Measurements

The overall porosities of the TFC FO membranes were measured via gravimetry method. The membranes were firstly cut into 3 cm × 3 cm sections and immersed in a water bath at successive temperatures of 30, 35 and 40 °C. Each temperature was performed for 24 h. Next, excess water was removed from the wetted membranes, and the membranes were weighed. The membranes were then dried in the oven for at least 24 h and weighed periodically until their weights were constant. The overall porosity of a membrane was calculated as:(19)$$\varepsilon =\frac{\frac{\left(W_{wet}-W_{dry}\right)}{\rho_{water,T}}}{\frac{W_{dry}}{\rho_{membrane}}+\frac{W_{wet}-W_{dry}}{\rho_{water,T}}}$$
where *W_wet_* and *W_dry_* are the weights of the wet and dry membrane (g), $\rho_{water,T}$ represents the pure water density at temperature, *T*, and $\rho_{membrane}$ represents the bulk density of the membrane (g cm^−3^).

#### 3.3.4. Fourier Transform Infrared Spectroscopy (FTIR)

The presence of chemical structures in the pristine support membranes and the functional groups of the TFC FO membranes were determined via attenuated total reflection-Fourier transform infrared spectroscopy analysis (ATR–FTIR) using a Spectrum 65 FT-IR Spectrometer (Massachusetts, MA, USA) at room temperature in the range of 450–4000 cm^−1^. Prior to ATR-FTIR characterization, 1 cm×1 cm samples from each support and its resulting TFC FO membrane were dried at room temperature overnight.

#### 3.3.5. Reverse Osmosis (RO) Experiments

A lab-scale RO dead-end test unit (HP4750, Sterlitech Co., Kent, WA, USA) with an effective membrane area of 14.6 cm^2^ was used to determine the water permeability coefficient (*A*), salt rejection (*R*), and salt permeability coefficient (*B*) of each TFC FO membrane using deionized water (*A*) and 2 g L^−1^ NaCl solution (*R* and *B*). For the water permeability coefficient (*A*), pressure was applied at a constant increment of 4bar from 10 to 18 bar. The experimental water flux for each TFC FO membrane *J_w_* (L m^−2^ h^−1^ bar^−1^) was computed with:(20)$$J_{w}=\frac{\Delta V}{A_{m}\times \Delta t}$$
where Δ*V* is the volume of the permeate (L) collected over a recorded time period Δ*t* (h) and *A_m_* is the effective membrane area (m^2^). The water permeability (L m^−2^ h^−1^ bar^−1^) of the membrane was then computed as [3]:(21)$$A=\frac{J_{w}}{\Delta P}$$
where Δ*P* is the pressure difference. The *R* (%) value was determined by using 2 g L^−1^ of a NaCl solution under a pressure of 18 bar:(22)$$R=\left(1-\frac{C_{p}}{C_{f}}\right)\times 100\%,$$
where *C_p_* and *C_f_* are the bulk concentrations (g L^−1^) of the permeate and the FS, respectively. Finally, the solute permeability, *B* was determined with:(23)$$B=A\frac{\left(1-R\right)\left(\Delta P-\Delta \pi \right)}{R}$$

### 3.4. FO Experiments

#### Bench-Scale System

A flat-sheet modular bench-scale FO system, illustrated Figure 2, was used to obtain the FO (AL–FS orientation) water flux (Equation (20)) and reverse salt flux (RSF) (Equation (24)) of each TFC FO membranes at the three different temperatures. The RSF is described as solute permeability, *J_S_*:(24)$$J_{s}=\frac{\Delta \left(C_{t}\times V_{t}\right)}{A\times \Delta t}$$
where Δ*C_t_* (g L^−1^) and Δ*V_t_* (L) are the concentration and volume changes of the FS, respectively, at the end of the FO tests [3,26]. The effective membrane area was 42 cm^2^. The FS (deionized water) and DS (2.0 g L^−1^ NaCl) were prepared in volumes of 15.0 ± 0.1 L in two separate tanks at the beginning of the experiment. Both the FS and DS were circulated at 1.5 L min^−1^ in a counter-current closed loop driven by high-pressure pumps (Hydra-Cell).

A bypass temperature regulator with a controller linked to a computer was used to regulate the temperature of the DS. In the FO experimental setup, the DS tank was placed on an electronic balance connected to a computer for continuous weight measurements. The flux was calculated according to the weight changes in the tank. The RSF was based on the concentration differences over time in the FS measured with a conductivity meter. These experiments were stabilized for 30 min and run for 300 min in total.

## 4. Results and Discussion

### 4.1. Effect of the PSf:PES Ratio on PA Structure 

The substrates made with different PSf:PES ratios resulted in different polyamide (PA) structures under the constant interfacial polymerization fabrication conditions due to the thermodynamic stabilities of the dope solutions giving rise to different diffusion rates [16,27,28]. Figure 3 shows the SEM surfaces (left) and images of the cross sections (right) of the three TFC FO membranes. A thick and typical globular-like PA layer can obviously be seen on the MS1(a). However, for the MS3(c), a thinner and very non-globular-like PA layer was observed. Both S1 and S3, which represent pristine PSf and PES (less hydrophilic), respectively, created greater mass resistances, which hindered PPD diffusion [29]. Hence, in both cases, a large number of unreacted PPD molecules remained on the top surface of the support before the TMC was added, thus forming a thicker PA layer after interfacial polymerization is completed, resulting in excellent salt rejection [30,31]. Meanwhile, there is a great deal of smaller ridge-and-valley morphology on MS2′s (b) surface due to its hydrophilicity, which allows a higher PPD diffusion rate into the organic phase and has more participating monomers during the interfacial polymerization reaction [9].

All the membranes have homogenous finger-like porous structures underneath their surfaces and along their lengths. Finger-like structures in the support membrane are more favourable for molecular transport in the FO process because they reduce the mass resistance and *S* parameter of a membrane [32]. Some large, finger-like macrovoids can be clearly seen in the lower part of MS1(d) due to the delayed phase inversion, whereas the open-bottom structure of MS3(f) was probably formed by the fast withdrawal of the solvent during the phase inversion due to the high thermodynamic instability of the dope solution [20]. A homogenous finger-like porous structure without an open bottom was observed in MS2(e) because it is more thermodynamically stable than MS1 and MS3 as a result of blending the less-polar PES with the PSf. The overall thicknesses of the TFC FO membranes decrease from MS1 to MS3. The thickness of the films varied due to the different precipitation rates of the polymer solutions with different PSf:PES ratios [33].

### 4.2. Effect of the PSf:PES Ratio on the Overall Membrane Wettability

Table 2 shows the effects of the substrate ratios and interfacial polymerization reactions on the hydrophilicities of the TFC FO membranes. All the support membranes without PA layers were less hydrophilic and had higher CA measurements, namely, 81.9 ± 0.6°, 75.3 ± 1.7°, and 75.5 ± 0.4° for S1, S2, and S3, respectively, than their respective TFC FO membranes, which had CAs of 65.2 ± 5.4°, 64.5 ± 7.8°, and 66.6 ± 3.5°, respectively. This finding is due to the presence of hydroxyl and amide groups on the TFC FO membranes’ surface arising from the breakdown of the TMC groups, as illustrated in the FTIR analysis [10].

### 4.3. Effect of the PSf:PES Ratios on the Functional Group Analysis of the PA Layer 

The functional groups and their intensities for each membrane were examined by ATR–FTIR analysis, and the results are presented in Figure 4. Comparisons were made between TFC FO membranes and their corresponding supports. In general, no obvious peculiarities were visible in the diagrams, except for the significant change in the intensities of the O–H band in the 3000–3680 cm^−1^ region between the TFC FOs and their support membranes. It is postulated that the increases in intensity may have helped increase the surface hydrophilicities of the TFC FO membranes, as evidenced by the CA measurements, that is, MS2 > MS1 > MS3 [34,35]. The weak peaks observed at 1660 cm^−1^ and 1550 cm^−1^ in TFC FO membranes correspond to amide I (C = O stretching) and amide II (N–H bending), both due to the formation of product groups during the IP reaction. The weaker peaks appearing at 1129 cm^−1^ for the TFC FO membranes compared to their supports, especially in the cases of MS1 and MS2, correspond to the asymmetric stretching and deformation vibrations of sulfonic acid groups [36].

### 4.4. Effect of the PSf:PES Ratio on Membrane Porosity

The free volume describes the freedom of water molecules to move about in a volume of liquid, meaning that the molecules could be migrated thermally within a macroscopic porous structure [37]. Under ordinary conditions, the porosity increases with an increase in temperature. The overall porosities of each support membrane at different temperatures are shown in Figure 5. It was found that the overall porosities of all TFC FO membranes increased as the temperature increased from 30 to 40 °C in 5° increments. A similar outcome was obtained by Fabian et al. [38], that is, that porosity is non-proportionally linear to temperature and is thermally and externally adjustable.

The porosities of MS1, MS2, and MS3 increased from 53.82 ± 3.44 to 60.30 ± 1.17% (12.04%), 69.49 ± 1.32 to 80.05 ± 0.80% (15.20%), and 52.73 ± 2.54 to 61.37 ± 3.60% (16.39%), respectively. MS1 and MS3 showed similar characteristics in terms of their porosities at all three experimental temperatures due to their similar precipitation rates giving rise to similar porosities during phase inversion. MS2 was distinctly different when compared to MS1 (19.75% higher) and MS3 (18.68% higher). MS2 can retain more water than MS1 and MS3, giving it better wettability. At elevated temperatures closer to the glass transition temperature, the intermolecular chains in MS2 loosen and reserve more water in terms of free volume [39]. The CA measurements, ATR–FTIR and SEM analyses, and overall porosity measurements show that MS2 has better hydrophilicity than MS1 and MS3. Thus, it is believed that MS2 provides a good separation medium with better water permeability and a lower *S* parameter.

### 4.5. Determination of Water and Solute Permeability Coefficients A and B 

Figure 6 shows the water (*A*) and solute (*B*) permeability coefficients and their corresponding solute rejection (*R*) for the synthesized TFC FO membranes with supports made from different PSf:PES ratios. MS2 has both the highest *A* and *B* values at 3.79 ± 1.06 L m^−2^ h^−1^ bar^−1^ and 8.42 ± 2.34 g m^−2^ h^−1^, respectively. This result is in line with it having the most intense hydroxyl groups (–OH) and the highest surface roughness, which is postulated to attract more water molecules to its surface, as the roughness increases the contact area for water molecules. The surface hydrophilicity of MS2 provides good water and salts permeation through the large presence of hydrogen bonds and Van der Waals forces [40]. However, the rejection rate for MS2 (74%) was the lowest. Ghosh and Hoek [27] reported that large pore sizes in membranes increase water permeation and reduce salt rejection. However, the mean pore size is difficult to measure. The overall porosities of the membrane measured in the present study indicate that the porosity also affects water permeation and salt rejection.

MS1 has the highest solute rejection of 92%, lowest *B* value of 0.94 ± 0.23 g m^−2^ h^−1^, and a low *A* value of 1.71 ± 0.41 L m^−2^ h^−1^ bar^−1^. Compared to MS1 and MS2, MS3 has the lowest *A* value of 1.37 ± 0.21 L m^−2^ h^−1^ bar^−1^ in addition to a *B* of 1.89 ± 0.29 g m^−2^ h^−1^, and a solute rejection rate of 82%. The three membranes have similar porosities, but MS3’s higher free volume (open bottom) provides low mechanical strength at higher pressure, resulting in deformations and damage to the feed channel and the subsequent low water permeation and great salt blockage [41].

### 4.6. Effect of Temperature on the FO Performances of the TFC FO Membranes

Temperature plays a critical role in the hydrodynamics of solutions and membranes, especially in osmotically-driven membrane processes. Table 3 shows the characteristics of the 2 g L^−1^ NaCl solution at different temperatures. The density, kinematic viscosity, and mass resistance coefficients decrease but the bulk osmotic pressure, diffusivity, and mass transfer coefficients increase at elevated temperatures. These changes enhance the osmotic driving force in term of water flux. Nevertheless, dilutive internal concentration polarization (dICP) occurs when water permeation is high within the support membrane, leading it to resist the salt diffusion from the DS [42].

Figure 7 shows the FO water fluxes and RSFs of the TFC FO membranes at different temperatures. In the FO mode, the initial water fluxes obtained for all TFC FO membranes were high and decreased drastically over time. As presented in Figure 7a,c, the FO performance of MS1 was enhanced from 10.48 to 15.71 L m^−2^ h^−1^ (49.90%) when the temperature increased from 30 to 40 °C, whereas MS3’s performance increased from 8.57 to 12.38 L m^−2^ h^−1^ (44.46%) over the same span. However, MS2 exhibited a huge increase of more than 300% in its FO performance, going from 13.81 to 42.86 L m^−2^ h^−1^, see Figure 7b. The enhanced porosities of all membranes at elevated temperatures allow greater water permeation. However, a higher dICP effect, whereby the high initial water permeation at increased temperatures reduces the effective osmotic pressure gradient, which is also the effective osmotically-driven force, results in rapidly decreasing water fluxes at 35 and 40 °C. It should be noted that the water fluxes decreased gradually at 30 °C for all the membranes tested because, at this ambient temperature, there is a low fouling tendency in FO operations. Augmentation with the water from the FS decreases the concentration of the DS over time, resulting in a reduced effective osmotic pressure gradient between the FS and DS, consequently reducing the water flux with the smallest dICP effect.

To achieve osmotic pressure equilibrium in an FO process, salts from the DS move into FS in the direction opposite that of the water flux in what is known as the RSF. The RSFs are stable at 30 °C over time because, at this ambient temperature, the variations in the pore radii and distribution are small. The exchange of water and salts down the concentration gradient is thermally stable and can achieve equilibrium. As presented in Figure 7d,f, the RSFs for MS1 and MS3 decreased gradually and remained constant throughout the FO experiments at 35 and 40 °C. By elevating the temperature, the salt diffusivity *D* was significantly enhanced. More salts transfer across the microscopic structures of the membranes from the DS to the FS due to the ease of diffusion at higher temperatures. When the salts are entrapped and saturated within the free volumes, a greater mass resistance ensues, reducing the RSF. This occurrence can be observed at 40 °C. In Figure 7e, the RSF for MS2 increased throughout the FO experiments at 35 and 40 °C due to its intrinsic behaviour and high water permeability, allowing it to wash off the salts accumulating within its free volume, thus enhancing the dICP effect, which is not preferable in FO operations [42]. The elevated temperature increases the diffusivity of the salts, and they penetrate the less dense and resistant PA active layer of MS2, which ultimately increases MS2′s RSF.

### 4.7. Effect of Bulk Temperature on the Structural Parameter S 

Figure 8 shows the *S_int_* and *S_eff_* values for all the membranes at different temperatures. In general, the *S_eff_* values are higher than the *S_int_* values. Since *S_int_* was determined based on a morphological assumption, it is postulated that the assumed values of the variables considered in this equation were lower than the actual values, especially the membrane tortuosity. Compared to pristine PSf and PES, the blended PSf:PES substrate exhibited the lowest *S* values, with values of 184 ± 4μm and 152 ± 26 μm for *S_int_* and *S_eff_*, respectively. In addition, a decreasing trend with increasing temperature was seen in the *S_int_* values for all the membranes. For MS1, the *S_int_* values dropped from 268 ± 17 to 239 ± 5 μm with a temperature increase from 30 °C to 40 °C, which is about a 12.13% reduction. For MS3, the *S_int_* decreased from 237 ± 11 to 203 ± 12 μm (approximately 16.75%). The *S_int_* values for MS2 dropped from 184 ± 4 to 159 ± 2 μm, which is equivalent to a 15.72% reduction. The enhanced porosity could potentially have led to greater water permeation and lower mass resistance, leading to the decrease in the *S_int_* values. The anisotropic membrane properties have long been recognized as constant in many scientific researches because they are difficult to be measured. Therefore, constant *S_int_* which involved only membrane properties was computed. However, the correlation of *S_int_* with temperature in this study shows that *S* is not constant and is influenced by the operating conditions. 

From Figure 8b, it can be seen that the *S_eff_* values were significantly more membrane dependent than temperature dependent. The *S_eff_* does not seem to be greatly influenced by the operating temperature, with the exception of the *S_eff_* for MS2. MS1 has a consistent *S_eff_* value even at elevated temperatures (going from 363 ± 45 to 360 ± 40 μm). In the case of MS3, the *S_eff_* values increased slightly from 370 ± 32 to 381 ± 30 μm. The salt accumulation within the membrane’s substrate, or the “salt wall”, that was created by the higher salt diffusivity when the temperature increased has induced a higher degree of internal concentration polarization (ICP). It is postulated that this occurrence is due to the wide-open bottom structure of the membrane shown in Figure 3, which possibly created a concentrative ICP phenomenon to boost the *S* of the membrane in contrast to the dICP. Moreover, MS3 had a relatively low water flux due to the built-up of its “salt wall”, which it was unable to wash off. The low water flux reduced the effective osmotic pressure gradient and driving force across the membrane, contributing to higher *S_eff_* values at elevated temperatures. In contrast, for MS2, the *S_eff_* values were significantly reduced from 152 ± 26 to 120 ± 14 μm (a 26.67% reduction). Its high-water flux in the FO experiments is considered to be the main contributor to its low *S_eff_* values. Moreover, significant deviations were observed between the *S_int_* and *S_eff_* values in all the membranes. Similar observation was reported by Xie et al. [43] and Wang et al. [10]. The changes on *S_eff_* for TFC FO membranes at low operating temperature (i.e., 18 to 40 °C) are insignificant. This may indicate a negligible thermal-induced changes of membrane properties. However, a drastic drop of *S_eff_* for MS2 was observed which could be due to its significant increase of the overall porosities which allowed a higher water flux as the temperature increased. The *S_int_* model indicates that *S* is a temperature-dependent parameter. However, when *S_eff_* model is used, temperature has little effect on the *S* value. Large experimental errors were present when determining the *S_eff_* values for the membranes, even at the same temperature, and this finding is consistent with the study conducted by Cath et al. [17]. Large errors are more likely to be present when membranes with low water permeability, such as MS1 and MS3, are tested. 

## 5. Conclusions

The present work aimed to improve the intrinsic parameters of the TFC FO membrane, especially its *S* parameter, by using combinations of PSf and PES as membrane substrates at ratios of 5:0, 2:3, and 0:5, respectively. Different operating temperatures ranging from 30 to 40 °C were used. The experimental results showed that the substrate made of PSf:PES was able to improve the membrane’s characteristics significantly with respect to surface hydrophilicity, overall porosity, and water permeability. Improved water permeability was observed in RO experiments and even in FO experiments. The MS2 membrane with the PSf:PES blend as a support membrane exhibited the highest water and salt permeabilities at 3.79 ± 1.06 L m^−2^ h^−1^ bar^−1^ and 8.42 ± 2.34 g m^−2^ h^−1^, respectively, and recorded a moderate salt rejection rate of 74%. In addition to altering the membrane’s characteristics, an analysis involving both *S* computation models was conducted by varying the temperature in gravimetry methods and through FO experiments and suggested that *S* is generally a membrane-dependent parameter and becomes a temperature-dependent parameter when the *S_int_* model is used. Compared to the other synthesized membranes, MS2 was the best membrane because its *S_int_* and *S_eff_* values were the lowest. The membrane’s characterization suggests that its high hydrophilicity due to its low CA, high porosity, and the high intensity of its hydroxyl group makes MS2 the preferred membrane. Since the industrial membrane operations required high operating pressure and temperature, the observation of membrane performances at elevated solution temperature in this study suggested that reduced macrovoids at the lower part of the membrane should be promoted. This could help to extend the shelf-life of the membranes although they may result in enhanced S value. Furthermore, it has the potential to meet the requirements of an FO membrane in future applications.

## Figures and Tables

**Figure 1 membranes-11-00448-f001:**
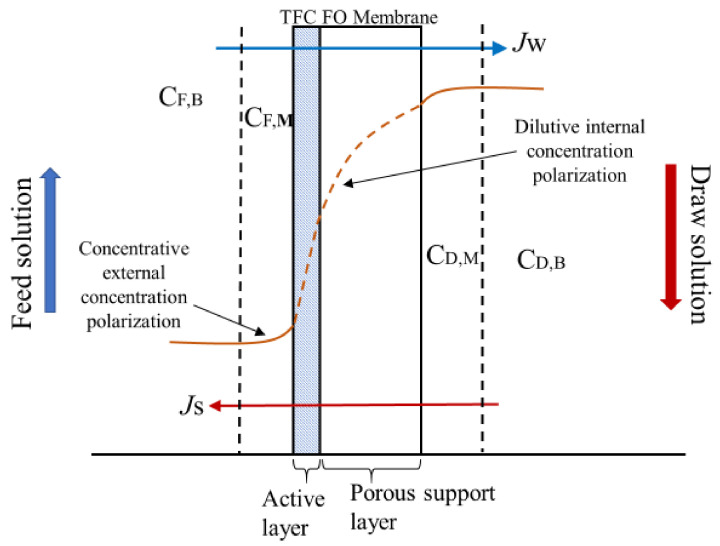
Forward osmosis (FO) concentration polarization phenomena.

**Figure 2 membranes-11-00448-f002:**
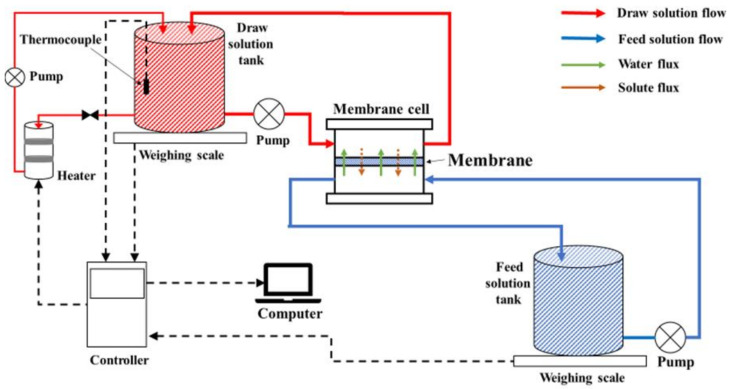
FO bench-scale flat-sheet module.

**Figure 3 membranes-11-00448-f003:**
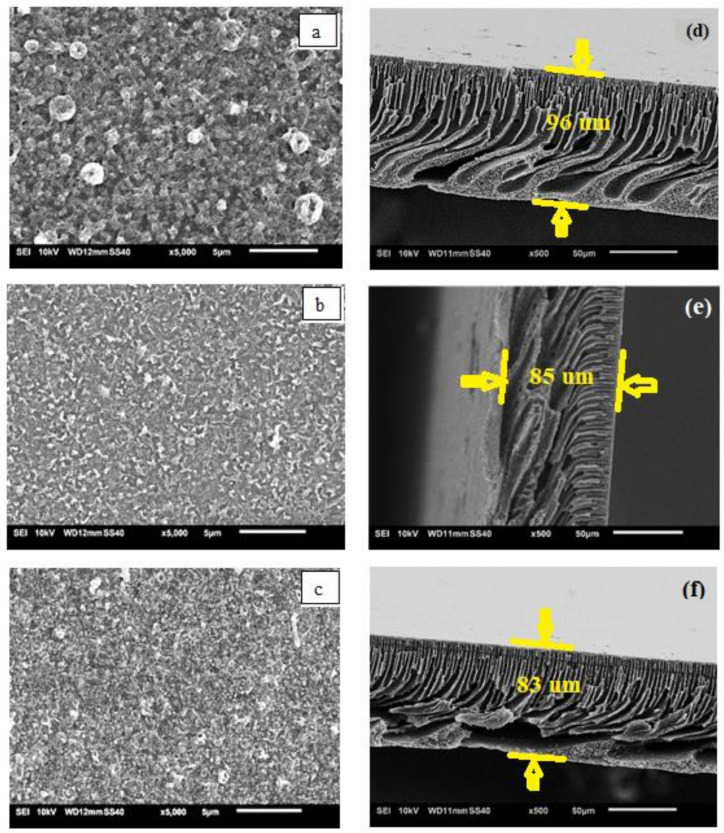
Scanning electronic micrography (SEM) top surfaces (**a**–**c**) and their respective cross sections (**d**–**f**) for the three TFC FO membranes.

**Figure 4 membranes-11-00448-f004:**
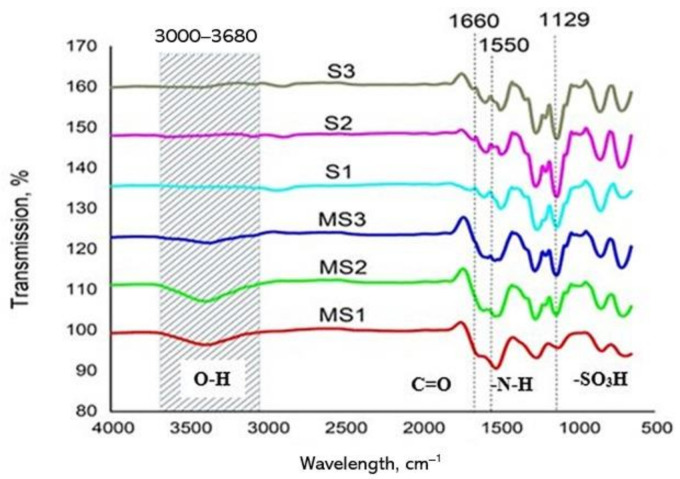
Attenuated total reflection–Fourier transform infrared spectroscopy (ATR–FTIR) Spectra of the synthesized TFC FO membranes.

**Figure 5 membranes-11-00448-f005:**
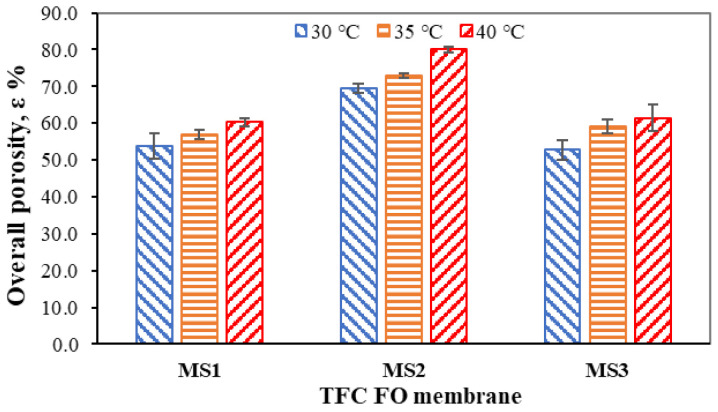
Overall porosities of the TFC FO membranes.

**Figure 6 membranes-11-00448-f006:**
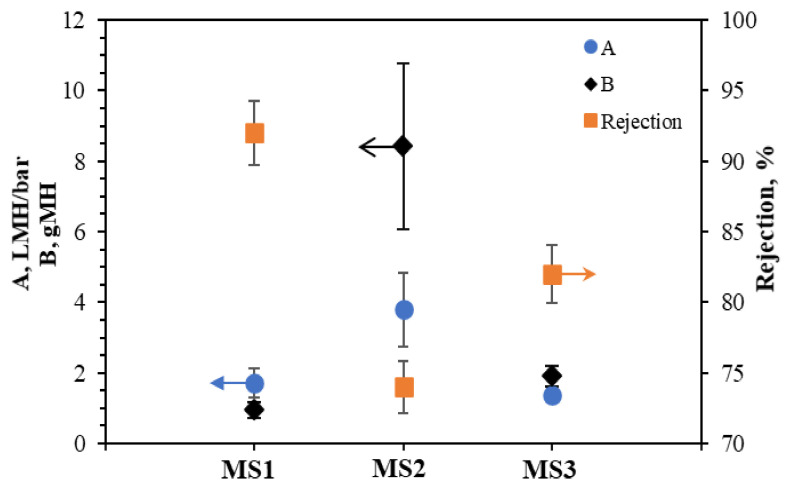
Water and salt permeability coefficients *A* and *B* and salt rejection (*R*) for the TFC FO membranes.

**Figure 7 membranes-11-00448-f007:**
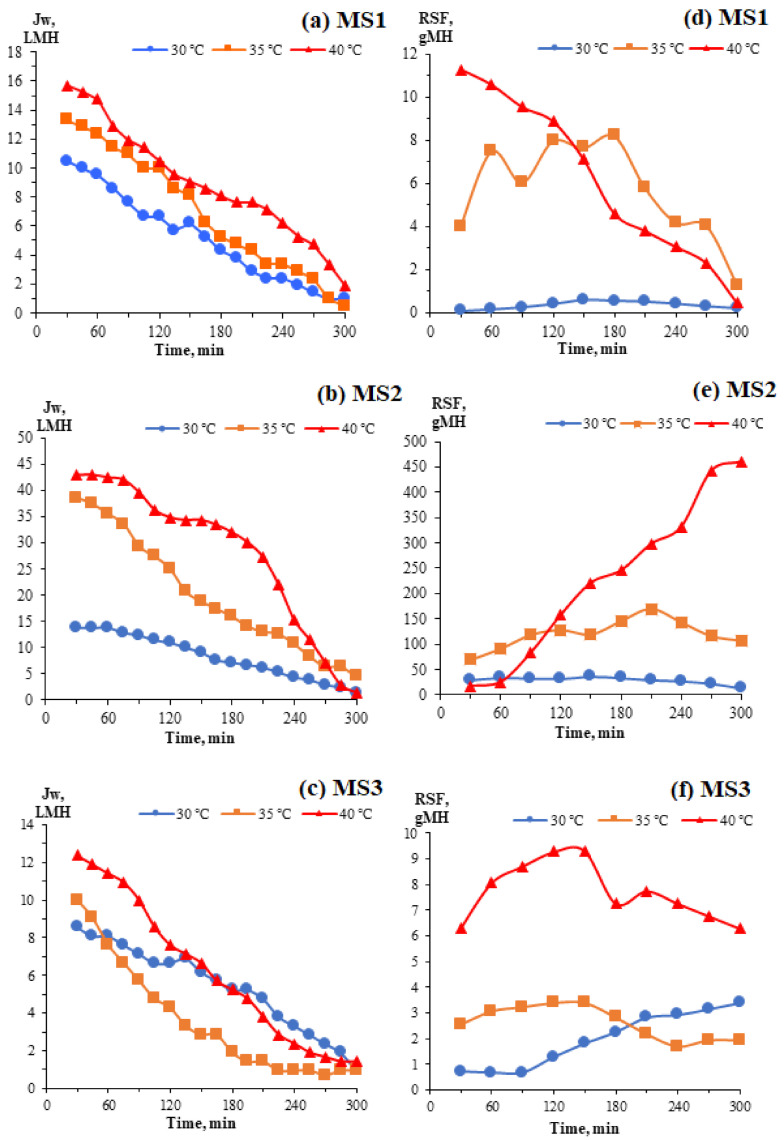
FO water fluxes (**a**–**c**) and RSFs (**d**–**f**) of the TFC FO membranes at different temperatures.

**Figure 8 membranes-11-00448-f008:**
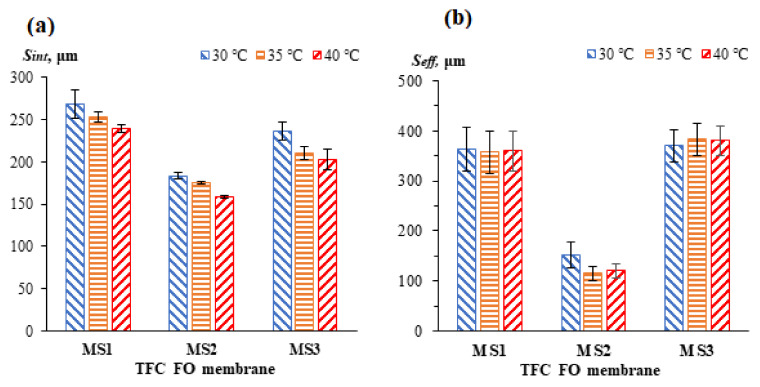
The (**a**) *S_int_* and (**b**) *S_eff_* values for the TFC FO membranes at different temperatures.

**Table 1 membranes-11-00448-t001:** Polymers compositions of the support membranes dissolved in N-methyl-2-pyrrolidone (NMP) solvent with 10 wt.% polyethylene glycol (PEG) 600.

Substrate Label	TFC FO Label	PSf (wt.%)	PES (wt.%)
S1	MS1	18	0
S2	MS2	10.8	7.2
S3	MS3	0	18

TFC: thin film composite; FO: forward osmosis; PSf: polysulfone; PES: polyethersulfone.

**Table 2 membranes-11-00448-t002:** Contact angles for the support membranes and their respective TFC FO membranes.

	S1	MS1	S2	MS2	S3	MS3
Contact angle (°)	81.9 ± 0.6	65.2 ± 5.4	75.3 ± 1.7	64.5 ± 7.8	75.5 ± 0.4	66.6 ± 3.5

**Table 3 membranes-11-00448-t003:** Characteristics of the 2 g L^−1^ NaCl solution at different temperatures.

Temperature, °C	30	35	40
Density *ρ*	997.2	995.6	993.7
Osmotic pressure *π*, in bar	1.713	1.741	1.769
Kinematic viscosity $\mu_{\mathrm{salt}\;\mathrm{water}\;}{\mathrm{in}\;m}^{2}{\;s}^{-1}$	$8.08\times 10^{-7}$	$7.27\times 10^{-7}$	$6.62\times 10^{-7}$
Diffusivity *D*	$2.74\times 10^{-9}$	$3.05\times 10^{-9}$	$3.36\times 10^{-9}$
Mass transfer *k*, in m s^−1^	$1.11\times 10^{-5}$	$1.19\times 10^{-5}$	$1.27\times 10^{-5}$
Mass resistance *K*, in s m^−1^	$6.34\times 10^{4}$	$4.89\times 10^{4}$	$3.69\times 10^{4}$

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
