# Peer review of "Improving the Structural Parameter of the Membrane Sublayer for Enhanced Forward Osmosis"

_membranes, 2021, doi:10.3390/membranes11060448_

Round 1

Reviewer 1 Report

The authors present an interesting manuscript which includes the fabrication via the phase inversion technique of a membrane with different polysulfone and poyethersulfone blend ratios as a substrate and using polyamide as an active layer. This work investigates the effect of the membrane substrate composition on the structural parameter (S) in the application of reverse osmosis. The experimental plan is well design and includes the evaluation of performance of the three synthetized membranes (MS1, MS2 and MS3) in lab-scale reverse osmosis experiments at temperatures of 30, 35 and 40ºC, and intrinsic and effective structural parameter are compared for the different operational conditions. A complete characterization of the membrane with different membrane substrate composition supports the experimental results of the RO operation [membrane functional groups and structure (via FTIR and SEM analysis), wettability (contact angle), porosity, and water and salt permeability and salt rejection rate].

The paper presents a good experimental work, well justified, contextualized, and described in the Materials and Methods Section. Discussion of the results is based on a revision of the state of the art, and it is well documented. This manuscript fits well withing the scope of this journal.

My consideration is that the quality of the paper is good enough to be published in this journal after some minor revisions:

  • A nomenclature list is recommended to better follow up of the article.
  • Results in Section 4.6 (Effect of the temperature on the FO performance) should be more contrasted with previous reported results, especially the last two paragraphs in which the Sint and Seff values are presented and discussed.

Author Response

The response to the reviewer's comment is as per attached.

Reviewer 2 Report

Sark et. al. by the study  presented in this paper, advanced the knowledge about polymer membrane systems. The deep characterization allowed the authors to evaluate and compare the structural parameter between the three different membranes realized and observed morphologically by SEM, chemically by ATR- FTIR, by contact angle measurement for wettability and respect their porosity feature. Reverse Osmosis experiments unveiled water permeability and then salt rejection. Permeation experiments were performed changing temperature, to evaluate how parameters can be affected by temperature at 30°C and 40 °C. In conclusion I think that this work is well done and exposed, graphical contribution is clear, formulas support adequately the experimental section, so this paper  merit to be published in the present form.

Just one observation, regard the statement from line 258 to 261, related to the macro voids presence, citing literature it is reported that macro voids enhance the structural parameter. However, as confirmed by the experiment result description in the section “Results and discussion” the deformation of the membrane applying pressure is worst where macro voids are larger, so in conclusion in my opinion is opportune to underline that macro voids have to be avoided and this is not well established in the conclusion section.

Author Response

(The authors gave the same response as above.)
